# Towards Flexible Inductive Bias via Progressive Reparameterization Scheduling

**Abstract.** There are two de-facto standard architectures in recent computer vision: Convolutional Neural Networks (CNNs) and Vision Transformers (ViTs). Strong inductive biases of convolutions help the model learn sample effectively, but such strong biases also limit the upper bound of CNNs when sufficient data are available. On the contrary, ViT is inferior to CNNs for small data but superior for sufficient data. Recent approaches attempt to combine the strengths of these two architectures. However, we show these approaches overlook that the optimal inductive bias also changes according to the target data scale changes by comparing various models' accuracy on subsets of sampled ImageNet at different ratios. In addition, through Fourier analysis of feature maps, the model's response patterns according to signal frequency changes, we observe which inductive bias is advantageous for each data scale. The more convolution-like inductive bias is included in the model, the smaller the data scale is required where the ViT-like model outperforms the ResNet performance. To obtain a model with flexible inductive bias on the data scale, we show reparameterization can interpolate inductive bias between convolution and self-attention. By adjusting the number of epochs the model stays in the convolution, we show that reparameterization from convolution to self-attention interpolates the Fourier analysis pattern between CNNs and ViTs. Adapting these findings, we propose Progressive Reparameterization Scheduling (PRS), in which reparameterization adjusts the required amount of convolution-like or self-attention-like inductive bias per layer. For small-scale datasets, our PRS performs reparameterization from convolution to self-attention linearly faster at the late stage layer. PRS outperformed previous studies on the small-scale dataset, e.g., CIFAR-100.

**Keywords:** Flexible Architecture, Vision Transformer, Convolution, Self-attention, Inductive Bias

## 1 Introduction

Architecture advances have enhanced the performance of various tasks in computer vision by improving backbone networks [3, 15, 16, 27, 28]. From the success of Transformers in natural language processing [2, 10, 31], Vision Transformers (ViTs) show that it can outperform Convolutional Neural Networks (CNNs) and its variants have led to architectural advances [22, 30, 36]. ViTs lack inductive

bias such as translation equivariance and locality compared to CNNs. Therefore, ViTs with sufficient training data can outperform CNNs, but ViTs with small data perform worse than CNNs.

To deal with the data-hungry problem, several works try to inject convolution-like inductive bias into ViTs. The straightforward approaches use convolutions to aid tokenization of an input image [14, 32–34] or design the modules [6, 12, 20, 35] for improving ViTs with the inductive bias of CNNs. Other approaches use the local attention mechanisms for introducing locality to ViTs [13, 22], which attend to the neighbor elements and improve the local extraction ability of global attention mechanisms. These approaches can design architectures that leverage the strength of CNNs and ViTs and can alleviate the data-hungry problem at some data scale that their work target.

However, we show these approaches overlook that the optimal inductive bias also changes according to the target data scale by comparing various models' accuracy on subsets of sampled ImageNet at different ratios. If trained on the excessively tiny dataset, recent ViT variants still show lower accuracy than ResNet, and on the full ImageNet scale, all ViT variants outperform ResNet. Inspired by Park *et al.* [24], we perform Fourier analysis on these models to further analyze inductive biases in the architecture. We observe that ViTs injected convolution-like inductive bias show frequency characteristics between it of ResNet and ViT. In this experiment, the more convolution-like inductive bias is included, the smaller the data scale is required where the model outperforms the ResNet performance. Specifically, their frequency characteristics tend to serve as the high-pass filter in early layers and as more low-pass filter closer to the last layer. Nevertheless, such a fixed architecture in previous approaches has a fixed inductive bias between CNNs and ViTs, making it difficult to design an architecture that performs well on various data scales. Therefore, each time a new target dataset is given, the optimal inductive bias required changes, so each time the model's architectural design needs to be renewed. For example, a CNN-like architecture should be used for small-scale dataset such as CIFAR [17], and a ViT-like architecture should be designed for large-scale dataset such as JFT [26]. Also, this design process requires multiple training for tuning the inductive bias of model, which is time consuming.

In this paper, we confirm the possibility of reparameterization technique [5, 19] from convolution to self-attention towards flexible inductive bias between convolution and self-attention during a single training trial. The reparameterization technique can change the learned convolution layer to self-attention, which identically operates like learned convolution. Performing Fourier analysis, we show that reparameterization can interpolate the inductive biases between convolution and self-attention by adjusting the moment of reparameterization during training. We observe that more training with convolutions than with self-attention makes the model have a similar frequency characteristic to CNN and vice versa. This observation shows that adjusting the schedule of reparameterization can interpolate between the inductive bias of CNNs and ViTs.

Table 1: **Comparison of various architectures** ✓ means that the model has the corresponding characteristics, and ✗ does not. ✓* indicates that ConViT's convolutional operation is given only in the initial training stage and then learned in the form of gated self-attention.

|  | DeiT [29] | ResNet [16] | ConViT [12] | ResT [35] | Swin [22] |
|---|---|---|---|---|---|
| Hierarchical Structure | ✗ | ✓ | ✗ | ✓ | ✓ |
| Relative Positional Encoding | ✗ | ✗ | ✓ | ✗ | ✓ |
| Local Attention | ✗ | ✗ | ✗ | ✗ | ✓ |
| Convolutional Operation | ✗ | ✓ | ✓* | ✓ | ✗ |

From these observations, we propose the Progressive Reparameterization Scheduling (PRS). PRS is to sequentially reparameterize from the last layer to the first layer. Layers closer to the last layers are more trained with self-attention than convolution, making them closer to self-attention. Therefore, we can make the model have a suitable inductive bias for small-scale data with our schedule. We validate the effectiveness of PRS with experiments on the CIFAR-100 dataset.

Our contributions are summarized as follows:

- We observe that architecture with a more convolutional inductive bias in the early stage layers is advantageous on a small data scale. However, if the data scale is large, it is advantageous to have a self-attentional inductive bias.
- We show that adjusting the remaining period as convolution before reparameterization can interpolate the inductive bias between convolution and self-attention.
- Based on observations of favorable conditions in small-scale datasets, we propose the Progressive Reparameterization Scheduling (PRS) which sequentially changes convolution to self-attention from the last layer to the first layer. PRS outperformed previous approaches on the small-scale dataset, e.g., CIFAR-100.

## 2    Related Work

### 2.1    Convolution Neural Networks

CNNs, the most representative models in computer vision, have evolved over decades from LeNeT [18] to ResNet [16] in a way that is faster and more accurate. CNNs can effectively capture low-level features of images through inductive biases which are locality and translation invariance. However, CNNs have a weakness in capturing global information due to their limited receptive field.

## 2.2  Vision Transformers

Despite the great success of vision transformer [11] in computer vision, ViT has several fatal limitations that it requires high cost and is difficult to extract the low-level features which contain fundamental structures, so that it shows inferior performance than CNNs in small data scales. There are several attempts to overcome the limitations of ViT and improve its performance by injecting a convolution inductive bias into the Transformer.

DeiT [29] allows ViT to take the knowledge of convolution through distillation token. They can converge a model, which fails in ViT. On the other hand, The straightforward approaches [4, 20, 34, 35] employ inductive bias to augment ViT by adding depthwise convolution to the FFN of the Transformer. ConViT [12] presents a new form of self-attention(SA) called Gated positional self-attention (GPSA) that can be initialized as a convolution layer. After being initialized as convolution only at the start of learning, ConViT learns only in the form of self-attention. Thus, it does not give sufficient inductive bias on small resources. Swin Transformer [22] imposes a bias for the locality to ViT in a way that limits the receptive field by local attention mechanisms. A brief comparison of these methods is shown in Table 1.

## 2.3  Vision Transformers and Convolutions

There have been several studies analyzing the difference between CNNs and ViTs [24, 25]. Park *et al.* [24] and Raghu *et al.* [25] prove that CNN and Transformer extract entirely different visual representations. In particular, Park *et al.* [24] present the several analysis of self-attention and convolution that self-attention acts as a low-pass filter while convolution acts as a high pass filter. Furthermore, several approaches [5, 8, 19] have reparameterized convolution to self-attention by proving that their operations can be substituted for each other. Cordonnier *et al.* [5] demonstrates that self-attention and convolution can have the same operation when relative positional encoding and the particular settings are applied. T-CNN [8] presents the model using GPSA proposed by ConViT, which reparameterizes convolution layer as GPSA layers. C-MHSA [19] prove that reparameterization between two models is also possible even when the input was patch unit, and propose a two-phase training model, which initializes ViT from a well-trained CNN utilizing the construction in above theoretical proof.

## 3  Preliminaries

Here, we recall the mathematical definitions of multi-head self-attention and convolution to help understand the next section. Then, we briefly introduce the background of reparameterization from convolution layer to self-attention layer. We follow the notation in [5].

**convolution layer** Convolution layer has locality and translation equivariance characteristics, which are useful inductive biases in many vision tasks. Those inductive biases are encoded in the model through parameter sharing and local information aggregation. Thanks to the inductive biases, better performance can be obtained with a low data regime compared to a transformer that uses a global receptive field. The output of the convolution layer can be roughly formulated as follows:

$$\text{Conv}(\boldsymbol{X}) = \sum_{\Delta} \boldsymbol{X}\,\boldsymbol{W}^C, \tag{1}$$

where $\boldsymbol{X} \in \mathbb{R}^{H \times W \times C}$ is an image tensor, $H,W,C$ is the image height, width and channel, $\boldsymbol{W}^C$ is convolution filter weight and the set

$$\Delta = \left[ -\left\lfloor \frac{K}{2} \right\rfloor, \cdots, \left\lfloor \frac{K}{2} \right\rfloor \right] \times \left[ -\left\lfloor \frac{K}{2} \right\rfloor, \cdots, \left\lfloor \frac{K}{2} \right\rfloor \right] \tag{2}$$

is the receptive field with $K \times K$ kernel.

**Multi-head Self-Attention Mechanism** Multi-head self-attention(MHSA) mechanism [31] trains the model to find semantic meaning by finding associations among a total of $N$ elements using query $\boldsymbol{Q} \in \mathbb{R}^{N \times d_H}$, key $\boldsymbol{K} \in \mathbb{R}^{N \times d_H}$, and value $\boldsymbol{V} \in \mathbb{R}^{N \times d_H}$, where $d_H$ is the size of each head. After embedding the sequence $\boldsymbol{X} \in \mathbb{R}^{N \times d}$ as a query and key using $\boldsymbol{W}^Q \in \mathbb{R}^{d \times d_H}$ and $\boldsymbol{W}^K \in \mathbb{R}^{d \times d_H}$, an attention score $\boldsymbol{A} \in \mathbb{R}^{N \times N}$ can be obtained by applying softmax to the value obtained by inner producting $\boldsymbol{Q}$ and $\boldsymbol{K}$, where $d$ is the size of an input token. Self-attention(SA) is obtained through matrix multiplication of $\boldsymbol{V}$ embedded by $\boldsymbol{W}^V \in \mathbb{R}^{N \times d_H}$ and $\boldsymbol{A}$:

$$\text{SA}(\boldsymbol{X}) = \boldsymbol{A}(\boldsymbol{X}\boldsymbol{W}^Q, \boldsymbol{X}\boldsymbol{W}^K)\boldsymbol{X}\boldsymbol{W}^V,$$
$$\boldsymbol{A}(\mathbf{Q}, \mathbf{K}) = \text{softmax}\left( \frac{\boldsymbol{Q}\boldsymbol{K}^\top}{\sqrt{d}} + \boldsymbol{B} \right), \tag{3}$$

where $\boldsymbol{B}$ is a relative position suggested in [7]. By properly setting the relative positional embedding $\boldsymbol{B}$, we can force the query pixel to focus on only one key pixel. MHSA allows the model to attend information from different representation subspaces by performing an attention function in parallel using multiple heads. MHSA with total of $N_H$ heads can be formulated as follows:

$$\text{MHSA}(\boldsymbol{X}) = \sum_{k=1}^{N_H} \text{SA}_k(\boldsymbol{X})\,\boldsymbol{W}_k^O, \tag{4}$$

where $\boldsymbol{W}^O$ is learnable projection and $k$ is the index of the head.

**Reparameterizing MHSA into Convolution Layer** [19] showed that $K \times K$ kernels can be performed through $K^2$ heads, where $K$ is the size of the kernel. Since the convolution layer is agnostic to the context of the input, it is necessary to set $\boldsymbol{W}^Q$ and $\boldsymbol{W}^K$ as $\boldsymbol{0}$ to convert the convolution to MHSA. Using equations (3) and (4) together, MHSA can be formulated as follows:

$$\text{MHSA}(\boldsymbol{X}) = \sum_{k=1}^{N_H} \boldsymbol{A}_k \boldsymbol{X} \, \boldsymbol{W}_k^V \, \boldsymbol{W}_k^O. \tag{5}$$

As $\boldsymbol{A}_k \boldsymbol{X}$ is used to select the desired pixel, the knowledge of the convolution layer can be completely transferred to the MHSA by setting $\boldsymbol{W}^V$ to $\boldsymbol{I}$ and initializing $\boldsymbol{W}^O$ to $\boldsymbol{W}^C$.

## 4    Inductive Bias Analysis of Various Architectures

In this section, we analyze various architectures through Fourier analysis and accuracy tendency according to data scale. Previous works designing the modules by mixing convolution-like inductive bias to ViTs overlook that a fixed architecture has a fixed inductive bias and optimal inductive bias can change according to data scale. To confirm it, we conducted the experiments which measure the accuracy of various architecture by changing data scale of ImageNet [9]. In these experiments, we observe that the required data-scale for outperforming ResNet is different for each architecture.

Then, we link frequency characteristics of the recent ViT variants and tendency of their accuracy with data scale by expanding observations of Park *et al.* [24]. With Fourier analysis of Park *et al.* [24], we observe that architecture having more CNN-like frequency characteristics shows CNN-like efficiency and accuracy tendency in the small-scale datasets.

### 4.1    Our Hypothesis

We hypothesize that 1) the more convolution-like inductive bias is included, the smaller the data scale is required where the ViT-like model outperforms CNNs and 2) frequency characteristics can explain whether the inductive bias of model is closer to CNNs or ViTs. Specifically, the incapacity to which the layer amplifies the high-frequency signal tends to dramatically increase from the first layer to last layer in CNN, whereas ViT does not increase well. ViTs injected with the inductive bias of convolutions tend to increase it, but not as drastic as CNN. Here, we observe that ViTs increasing this incapacity more dramatically perform well on smaller scale data like CNNs.

### 4.2    Data Scale Experiment

CNNs have the inductive biases such as locality and translation invariance and ViTs do not. Because of the difference in inductive bias that architecture has, the

Table 2: **Data scale experiment of various model architectures.** For a fair comparison, the data augmentation and regulation techniques during the learning process of all experimental models followed those of DeiT. [29].

| Model | ImgNet Ratio | Acc@1 | Acc@5 | Flops | # params |
|---|---|---|---|---|---|
| DeiT-Ti [29] | 0.01 | 6.43 | 16.37 | | |
| | 0.05 | 24.82 | 46.40 | | |
| | 0.1 | 38.61 | 63.26 | 1.25G | 5M |
| | 0.5 | 67.03 | 88.11 | | |
| | **1** | **72.2** | **91.1** | | |
| ConViT-Ti [12] | 0.01 | 6.08 | 15.82 | | |
| | 0.05 | 26.93 | 49.86 | | |
| | 0.1 | 42.92 | 67.78 | 1G | 6M |
| | 0.5 | 68.21 | 88.93 | | |
| | **1** | **73.1** | **91.7** | | |
| ResTv1-Lite [35] | 0.01 | 11.19 | 26.542 | | |
| | **0.05** | **42.92** | **67.91** | | |
| | **0.1** | **52.88** | **76.62** | 1.4G | 11M |
| | **0.5** | **73.03** | **91.39** | | |
| | **1** | **77.0** | **93.6** | | |
| ResNet-18 [16] | **0.01** | **13.93** | **30.85** | | |
| | 0.05 | 42.04 | 67.58 | | |
| | 0.1 | 52.24 | 76.38 | 1.8G | 11.6M |
| | 0.5 | 66.38 | 87.30 | | |
| | 1 | 69.53 | 89.08 | | |
| Swin-T [22] | 0.01 | 13.20 | 27.39 | | |
| | 0.05 | 38.69 | 61.88 | | |
| | 0.1 | 53.46 | 75.57 | 4.5G | 28M |
| | **0.5** | **76.21** | **92.86** | | |
| | **1** | **81.2** | **95.5** | | |
| ResNet-50 [16] | **0.01** | **13.67** | **30.19** | | |
| | **0.05** | **46.82** | **70.85** | | |
| | **0.1** | **58.14** | **80.53** | 3.6G | 23.9M |
| | 0.5 | 75.23 | 92.42 | | |
| | 1 | 80.15 | 94.49 | | |

data scale determines their superiority. In small-scale data, CNNs outperform ViTs, and at some point, ViTs outperform CNNs as the data scale grows. ViT variants injected with the convolution-like inductive bias have stronger inductive bias compared to naïve ViT, and the amount of data required to outperform ResNet will be less than it. In this subsection, we identify accuracy trends and the amount of data required to outperform ResNet for various architectures by changing the data scale.

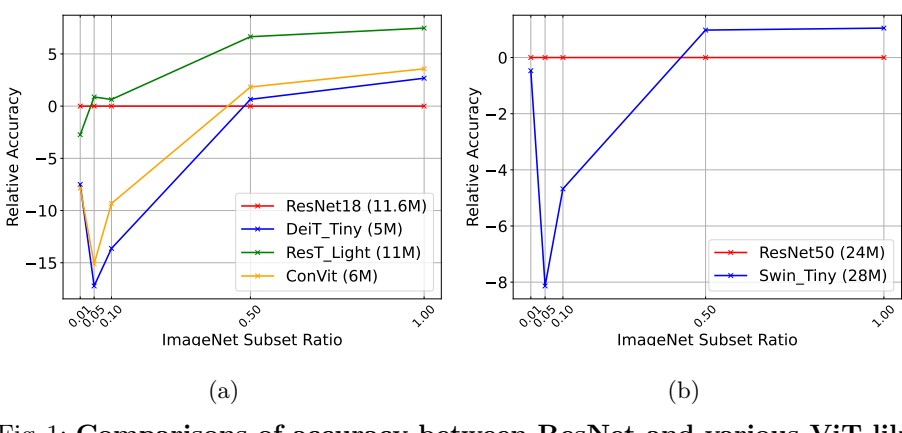

<p style="text-align:center">(a)                                    (b)</p>

Fig. 1: **Comparisons of accuracy between ResNet and various ViT-like architectures.** Each model trained on the subsets of imagenet, specifically 1%, 5%, 10%, 50% and 100%. We plot the accuracy difference between ResNet and other architectures with the increasing subset ratio. The numbers in parentheses mean the number of parameters of each model.

As shown in Table 2 and Figure 1, we make subsets with the ratio of 0.01, 0.05, 0.1, and 0.5 respectively in ImageNet for experiments in various settings with the same data distribution and different data scales. By utilizing the taxonomy of vision transformer proposed in [21], We choose the representatives in each category as ViT variants to compare together. ResT [35] injects inductive bias directly by adding convolution layers, whereas Swin [22] and ConViT [12] add locality in a new way. Swin uses a method that constrains global attention, while ConViT proposes a new self-attention layer that can act as a convolution layer in the initial stage of training. Therefore, we select ResNet-18 and ResNet-50 as the basic architecture of CNN, DeiT-Ti as Vanilla ViT and ResT-Light, ConViT-Ti, and Swin-T as the variations of the ViT to be tested. Since the number of parameters also significantly affects the performance, we compare the tiny version of Swin (Swin-T) [22] with ResNet-50 [16] and the remaining ViT variants with ResNet-18 [16]. Swin-T has more parameters than other models since the dimension is doubled every time it passes through one layer.

At 0.01, the smallest data scale, the ResNet series consisting of only CNNs shows better performance, and between them, ResNet-18 with smaller parameters has the highest accuracy. However, as the data scale increase, the accuracy of other ViT models increase more rapidly than ResNet. In particular, ResTv1-Light [35] and Swin-T [22], which have hierarchical structures, show superior performance among ViT variants and ResTv1-Light even records the highest accuracy of all models when the data scale is 0.05 or more.

As illustrated in Figure 1, DeiT-Ti [29] shows better performance than ResNet when the data scale is close to 1, while ConViT-Ti [12] and Swin-T [22] outperform it at 0.5 or more. meanwhile, the accuracy of ResT is higher than ResNet-18 from quite a small data scale of 0.05. Therefore, we argue that the inductive bias is strong in the order of ResTv1-Light, Swin-T, ConViT-Ti, and DeiT-Ti.

Through these experiments, we can prove that inductive bias and hierarchical structure have a great influence on accuracy improvement.

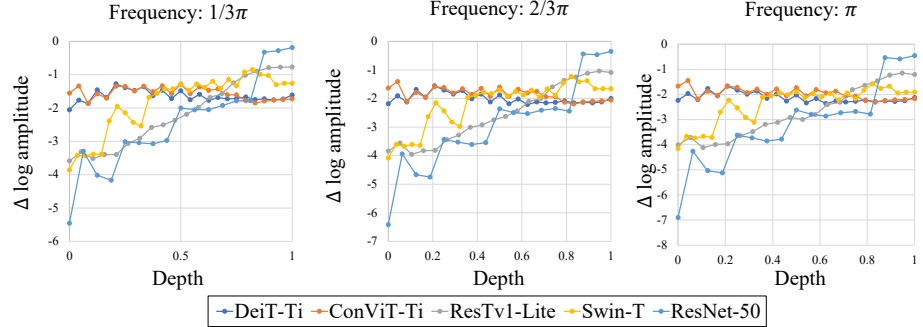

Fig. 2: **Frequency characteristics of ViTs and ResNet.** In ResNet-50, ResTv1-Lite, and Swin-T, the difference in log amplitude sharply increases as the normalized depth increase. On the other side, DeiT and ConViT which softly inject inductive biases into models do not have this tendency.

### 4.3   Fourier Analysis

As shown in Section 4.2, the required data scale for outperforming ResNet is different for each architecture. Inspired by the analysis of Park *et al.* [24], we show that the architectures with frequency characteristics more similar to ResNet tend to outperform ResNet at smaller data scales through Fourier analysis.

As in [23, 24], the feature maps of each layer can be converted to two-dimensional frequency domain with Fourier transform. Transformed feature maps can be represented on normalized frequency, which frequency is normalized to $[-\pi, \pi]$. The high-frequency components are represented at $-\pi$ and $\pi$ and the lowest frequency components are represented at 0. Then, we use the difference in log amplitude to report the amplitude ratio of high frequency to low-frequency components. For better visualization, differences in log amplitude between 0 and $1/3\pi$, 0 and $2/3\pi$, and 0 and $\pi$ are used to capture the overall frequency characteristics well.

Figure 2 shows frequency characteristics through Fourier analysis. In the ResNet results, the difference in log amplitude sharply increases as the normalized depth increases. This shows that early layers tend to amplify the high-frequency signal, and the tendency to amplify the high-frequency signal decreases sharply as closer to the last layer. However, DeiT and ConViT which softly inject inductive biases into models do not have this tendency and their frequency characteristics are similar through the layers. The results of Swin and ResT that strongly inject inductive biases into models with the local attention mechanism or convolution illustrate that the increase of the difference in log amplitude shows an intermediate level between it of ResNet and DeiT.

By combining the results of Figure 2 and Table 2, we can see that the model performs well for small-scale data if the increase in the difference in log amplitude through layers is sharp. It becomes smoother in the order of ResNet, ResT, Swin, ConViT, and DeiT, the accuracy is higher in the low-data regime in this order. These results are consistent with the observations of previous work that the inductive bias of CNNs helps the model to learn on small-scale data. From these, we address that the difference in log amplitude through the layers can measure the CNN-like inductive bias of the model. If it increases sharply similar to CNNs, the model has strong inductive biases and performs well in low-data regime.

## 5   Reparameterization Can Interpolate Inductive Biases

As shown on Section 4, a fixed architecture does not have flexible inductive bias, causing them to have be tuned for each data. Since modifying the architecture to have a suitable inductive bias for each data is too time-consuming, the method which can flexibly adjust the inductive bias during the training process is needed.

We observe that the model trained more with CNN than self-attention have more CNN-like frequency characteristics through reparameterization. With these results, we show that reparameterization can interpolate the inductive bias between CNNs and ViT by adjusting the moment of reparameterization during training.

### 5.1   Experimental Settings

Because reparameterization can change convolution to self-attention, we can adjust the ratio of epochs that each layer is trained with convolution and self-attention. In a 10% subset of the ImageNet data, we adjust this ratio by four settings: model trained with 1) convolution for 300 epochs and self-attention for 0 epochs, 2) convolution for 250 epochs and self-attention for 50 epochs 3) convolution for 150 epochs and self-attention for 150 epochs and 4) convolution for 50 epochs and self-attention for 250 epochs. We note that the model is more trained with convolution from 1) to 4). We follow the setting for reparameterization as in CMHSA-3 [19] and Fourier analysis as in Section 4.3.

### 5.2   Interpolation of Convolutional Inductive Bias

Figure 3 shows the results of Fourier analysis according to the ratio of trained epoch with convolution and self-attention. When comparing 1) to 4), we can see that the degree of increase become smaller from 1) to 4). As the ratio trained with self-attention increases, the difference in log amplitude of early stage layers tends to increase, and the difference in log amplitude of late stage layers tends to decrease. These results show that the more training with convolution make the degree of increase sharper. As we observed in the Section 4.3, the more sharply increasing the difference of log amplitude through normalized depth represents

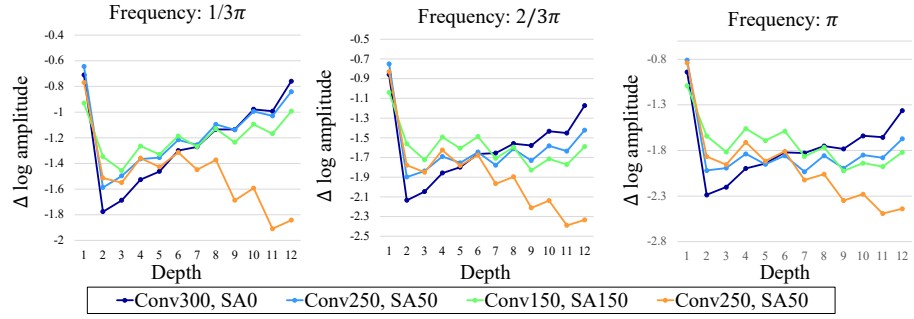

Fig. 3: **Visualization of Interpolation.** As the ratio trained with self-attention increases, the difference in log amplitude of early stage layers tends to increase, and the difference in log amplitude of late stage layers tends to decrease. Conv $x$, SA $y$ denotes that the model is trained with convolution for $x$ epochs and self-attention for $y$ epochs.

that the model have more CNN-like inductive biases. By combining the results of Figure 3 and this observation, we can see that the more trained with convolution make the model have more CNN-like inductive biases.

## 6  Progressive Reparameterization Scheduling

We now propose Progressive Reparameterization Scheduling (PRS) which adjusts the inductive bias of ViT for learning on small-scale data. PRS is based on our findings as:

- As shown in Section 4, the more convolution-like inductive bias is included, the smaller the data scale is required where the ViT-like model outperforms CNNs. In more detail, we can see that the model performs well for small-scale data if the increase in the difference of log amplitude through layers is sharp.
- Furthermore, in the interpolation experiment in Section 5, if the layer is trained in a convolution state for longer epochs, the layer has more convolution-like characteristics. If the layer is trained in a self-attention state for longer epochs, the layer has more self-attention-like characteristics. That is, by adjusting the schedule, it is possible to interpolate how much inductive bias the model will have between self-attention and convolution.

From these findings, PRS makes the early layer have a small difference in log amplitude as a high-pass filter and the last layer has a large difference in log amplitude as a low-pass filter. Because convolution and self-attention serve as high-pass filter and low-pass filter respectively as in Park *et al.* [24], PRS wants the rear layer to play the role of self-attention and the front layer to play the role of convolution. In order to force the rear layers to focus more on the role

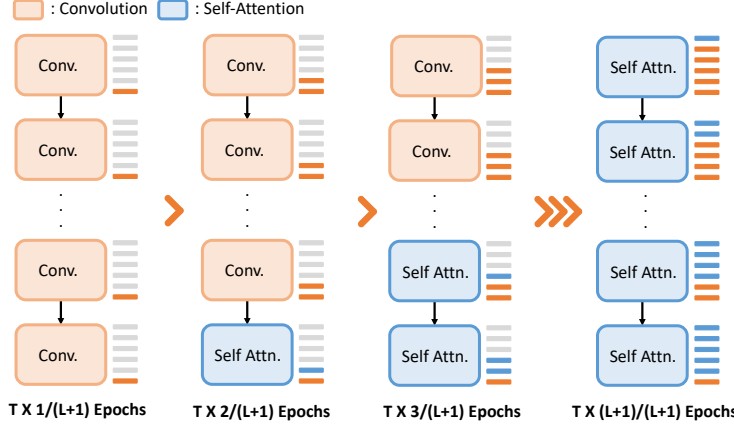

Fig. 4: **Illustration of PRS.** Conv. is a block with a convolutional layer, and Self Attn. is a block with a self-attention layer. Each block is progressively transformed from a convolution block to a self-attention block as the training progresses.

of self-attention than the front layers, PRS reparameterizes according to linear time scheduling from convolution to self-attention, starting from the rear part. PRS is depicted in Figure 4 and can be expressed as a formula as follows:

$$z_0 = \text{PE}(\boldsymbol{X}), \tag{6}$$

$$z'_l = \begin{cases} \text{Conv}(\text{LN}(\boldsymbol{z}_{l-1})) + \boldsymbol{z}_{l-1}, & (t \leq T \cdot (1 - \frac{l}{L+1})) \\ \text{MHSA}(\text{LN}(\boldsymbol{z}_{l-1})) + \boldsymbol{z}_{l-1}, & (t > T \cdot (1 - \frac{l}{L+1})) \end{cases} \tag{7}$$

$$z_l = \text{MLP}(\text{LN}(\boldsymbol{z}'_l)) + \boldsymbol{z}'_l, \tag{8}$$

$$\mathbf{y} = \text{Linear}(\text{GAP}(\boldsymbol{z}_L)), \tag{9}$$

where $\text{PE}(\cdot)$ is the patch embedding function that follows [19], $\text{LN}(\cdot)$ is LayerNorm [1], $\text{GAP}(\cdot)$ is global average pooling layer, $\text{Linear}(\cdot)$ is linear layer, $t$ denotes current epoch at training, $L$ denotes the total number of layers, $l = 1, 2, \cdots, L$ denotes the layer index and $T$ denotes the total number of training epochs, $\mathbf{y}$ denotes the output of the model.

Table 3 shows the effectiveness of PRS in CIFAR-100 dataset. PRS outperforms the baseline with a top-1 accuracy score of +2.37p on the CIFAR-100 dataset, showing that the performance can be boosted by a simple scheduling. We note that our PRS achieves better performance than the previous two-stage reparameterization strategy [19]. These results show that PRS can dynamically apply an appropriate inductive bias for each layer. Through the successful result of PRS, we conjecture that flexibly inducing inductive bias with reparameterization has the potential for designing the model on various scale data.

Table 3: **Training results of PRS.** We train the model for 400 epochs on the CIFAR-100 [17] dataset with our method, Progressive Reparameterization Scheduling.

| Model | Acc@1 | Acc@5 | Layers | #Heads | dim$_{emb}$ |
|---|---|---|---|---|---|
| ViT-base [11] | 60.90 | 86.66 | 12 | 12 | 768 |
| DeiT-small [29] | 71.83 | 90.99 | 12 | 6 | 384 |
| DeiT-base [29] | 69.98 | 88.91 | 12 | 12 | 768 |
| CMHSA-3 [19] | 76.72 | 93.74 | 6 | 9 | 768 |
| CMHSA-5 [19] | 78.74 | 94.40 | 6 | 9 | 768 |
| Ours w/CMHSA-3 | **79.09** | **94.86** | 6 | 9 | 768 |

## 7   Conclusion

From the analysis of existing ViT-variant models, we have the following conclusion: the more convolution-like inductive bias is included in the model, the smaller the data scale is required where the ViT-like model outperforms CNNs. Furthermore, we empirically show that reparameterization can interpolate inductive biases between convolution and self-attention by adjusting the moment of reparameterization during training. Through this empirical observation, we propose PRS, Progressive Reparameterization Scheduling, a flexible method that embeds the required amount of inductive bias for each layer. PRS outperforms existing approaches on the small-scale dataset, e.g., CIFAR-100.

**Limitations and Future Works** Although linear scheduling is performed in this paper, there is no guarantee that linear scheduling is optimal. Therefore, through subsequent experiments on scheduling, PRS can be improved by changing it to learnable rather than linearly or to another fancy method. In this paper, we only covered datasets with scales below ImageNet, but we will also proceed with an analysis of larger data scales than ImageNet. We also find that the hierarchical architectures tend to have more CNNs-like characteristics than the non-hierarchical architectures. This finding about hierarchy can further improve our inductive bias analysis and PRS.

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
