# OpenReview forum: "Towards Flexible Inductive Bias via Progressive Reparameterization Scheduling"
_thecvf.com/ECCV/2022/Workshop/VIPriors — VIPriors 2022 OralPosterTBD_

### Official Review · Reviewer_XLoE · 2022-07-20
**Good paper! Accept without doubt.**

**Rating:** 7
**Confidence:** 3

**Review:**

This paper first studies the inductive bias in convs and transformers using Fourier analysis. The amount of inductive bias is defined as the log-scale difference between the high and low frequencies. The observation seems to align with common sense that convs have much more bias thus being data-efficient, while transformers are the opposite.  Later, the authors study a progressive reparameterization schedule which allows a model to flexibly decide "optimal" the amount of bias needed during training by progressively converting convs to transformer blocks. In general, an interesting paper.

However, I do have a concern about the lack of connection among frequencies, inductive biases, and data efficiency. The main argument is that MSA is low-pass while convs are high-pass. However, it is not clear how the high/low frequency contributes to data efficiency.  Is the frequency bias universal (applicable to other datasets/tasks)? Reviewer "zxBu" shares the same opinion, as indicated in "The relationship between the frequency response of convolution/self-attention and their inductive bias is not properly explained and not obvious to me."

In general, I think the ProgressiveReparameterizationScheduling is an interesting idea and the experiments on cifar show some inspiring results.  Despite the concerns on the Fourier analysis, I would still vote for acceptance.

line 388 typo
line 389: how are the frequencies normalized to [-pi, pi]? linearly?
line 396: what is the 'normalized depth'?
is it possible to show some qualitative analysis/visual examples on low/high frequencies? what are the low/high frequencies exactly? How do they differ in early and later layers? It might be beneficial to understand the inductive bias.
fig 3 why is the orange line has the same configuration as the light blue line? is this a mistake? I would assume it is orange - conv100, SA200?

---

### Official Review · Reviewer_zxBu · 2022-07-22
**Interesting method, but concerns about motivating analysis**

**Rating:** 5
**Confidence:** 3

**Review:**

**Summary**
The authors use existing work that reparameterizes self-attention to be able to interpolate between self-attention and convolution, and propose a linear schedule that progressively switches over CNN modules to self-attention over training time. This method is motivated by a reproduction of the result that CNNs and ViT models with increased inductive biases are more data efficient than baseline ViT models, while ViT models can outperform CNNs at large data scales.

**Strengths**
- The proposed method is simple and easy to reproduce based on the paper.
- The method is effective on CIFAR-100.

**Weaknesses**
- In my view, the analysis in sections 4 and 5 does not connect to the method. The relationship between the frequency response of convolution/self-attention and their inductive bias is not properly explained and not obvious to me. The cited work by Park et al. doesn’t directly connect the two either. The claim that the frequency response is indicative of inductive bias is therefore, in my opinion, not substantiated.
- Regardless of the validity of the analysis in sections 4 and 5, in my opinion sections 4 and 5 are not necessary to argue that inductive bias aids data efficiency, as that is an established property of machine learning models. Specifically for convolutions and self-attention thee are many works to cite (see the related works in this work and the work of Park et al.). As such, the chapters distract from the main contributions.
- The main motivation for the method is improved accuracy on CIFAR-100, but a thorough description of the hyperparameter of the method and baselines is missing, which does not inspire trust in the results.
- The method is not evaluated for its claimed benefit of adjusting to the data scale. For example, the authors could have evaluated on the ImageNet subsets of sec 4.2.

Rating: marginally below acceptance threshold

**Justification**
My leading principle for this review is if I believe the paper disseminates useful information for the field. In the current form, I do not believe sections 4 and 5 should be published, as I do not agree with the claims on the relationship between frequency response and inductive bias, and they will confuse and distract uninformed readers. I can however see the value of the proposed linear schedule method, and think it would be a good fit for this workshop or another similar venue otherwise. I recommend the authors revisit the way they motivate and analyze their method, and resubmit to a similar venue.

---

### Official Review · Reviewer_kfrY · 2022-08-04
**Good paper that shows scientific significance, accept**

**Rating:** 7
**Confidence:** 4

**Review:**

**Summary** \
This paper investigates the inductive bias in ViT and CNN models. The authors claim that the previous design of injecting convolution-like inductive bias into CNN models ignore that the optimal inductive bias depends on the data scale and fixed inductive bias may not be optimal. Experiments on different data scales of ImageNet illustrate that smaller data scale is needed for ViT to outperform CNN if more convolution-like inductive biases is included. The paper also proves that frequency characteristics can explain whether the inductive bias is closer to convolution or self-attention by conducting Fourier analysis. Then the authors show that the interpolation of inductive bias between CNN and ViT can be realized by adjusting the moment of reparameterization during training. Based on the above findings, a progressively reparameterization scheduling is proposed to make the front layers to act like convolution and the rear layer to act like self-attention. Experiments on CIFAR-100 show the effectiveness of PRS.



**Strengths**
- Investigating the inductive bias injected in CNN and ViT models is crucial and this paper presents a new idea of making the inductive bias flexible, which provides a new research direction.
- The order of the paper and the logic in conducting this research is clear. The authors try to first understand the inductive bias by conducting the 'different data ratio' experiments and the Fourier analysis and then propose the scheduling strategy of reparameterization based on the previous findings.
- The tables, equations and figures used to demonstrate the PRS is clear.
- The paper is generally well-written.


**Weaknesses**
- The experiments used to verify the effectiveness of PRS is only conducted on CIFAR-100. It would be good if more datasets can be included.
- The legend in Figure 3 is misleadings. The orange line should be Conv50, SA250?

**Overall rating (1-10)** \
7

**Justification of rating** \
The scientific significance of this paper.

---

### Decision · Program_Chairs · 2022-08-08

**Decision:**

Accept (Oral/Poster TBD)

**Comment:**

Dear authors,


Congratulations! Your work has been accepted to the VIPriors workshop. Decisions on oral/poster presentations will follow later, when the program of the workshop is finalized.

*Please note the first action item is due on Wednesday! Please see instructions below.*

**Camera-ready instructions**

There is some work left to be done to ensure your work is included in the ECCV conference workshop proceedings. The ECCV publication managers use CMT to collect all workshop papers. This means we will migrate your paper from the VIPriors OpenReview page to the centralized ECCV workshop proceedings CMT page. The VIPriors program committee will ensure the details of your work (name, title, email address) are transferred to the CMT page, after which the ECCV proceeding managers will invite you to upload the camera-ready version of your work to the centralized ECCV CMT workshop proceedings page.

Please carefully follow the following instructions:
- **Before August 10th**, ensure that the first author has a CMT account under the same email address as the OpenReview account through which the accepted work was submitted. This account will be used to invite you to upload the camera-ready paper.
- Fill out this form, to inform us that the CMT account is in order: https://docs.google.com/forms/d/e/1FAIpQLSfyAoPv2_srESKaLRHIsHoWe3Fss1Z50ykdH7SzZpenA0m_5g/viewform
- Await instructions from the ECCV publication organizers, sent through CMT, on how to submit your camera-ready paper.
- Submit the camera-ready paper **before August 22nd**. Follow the camera-ready instructions for the main conference: https://eccv2022.ecva.net/submission/call-for-papers/.

**Attending the workshop**

We invite all authors of accepted works to attend the workshop in person on October 24th 2022 at ECCV in Tel Aviv. Please note a conference registration is required to attend the workshop. The workshop will be hybrid, enabling both in-person and remote attendance. We hope all accepted works can be represented in-person by at least one author, but we understand if this is not possible. Remote attendance of the workshop will be possible, though unfortunately there are limits on presenting works remotely: we intend to enable remote oral presentations, but this is not possible for posters.

Please fill out this form *before September 26th* to inform us of your attendance: https://docs.google.com/forms/d/e/1FAIpQLSfqRhdd2pq8t4CC8hL_c8fQo_TWcbzuQH3KGLzKVE36iTW_oQ/viewform.

**Presenting your work at the workshop**

Authors of all accepted papers are invited to present a poster at the workshop. Instructions on poster format will follow at a later date, but we will ask you to print and bring your own poster to the workshop.


For more information, as well as updates on the program of the workshop, keep an eye on our website: https://vipriors.github.io.

We thank you for choosing to submit to our workshop, and we are very much looking forward to hosting you in person in Tel Aviv!


Kind regards,

Robert-Jan Bruintjes
VIPriors program committee